# Comprehensive Review of Environmental Surveillance for Azole-Resistant *Aspergillus fumigatus*: A Practical Roadmap for Hospital Clinicians and Infection Control Teams

**DOI:** 10.3390/jof11020096

**Published:** 2025-01-25

**Authors:** Masato Tashiro, Yuichiro Nakano, Tomoyuki Shirahige, Satoshi Kakiuchi, Ayumi Fujita, Takeshi Tanaka, Takahiro Takazono, Koichi Izumikawa

**Affiliations:** 1Department of Infectious Diseases, Nagasaki University Graduate School of Biomedical Sciences, Nagasaki 852-8501, Japan; y-nakano@nagasaki-u.ac.jp (Y.N.); rtbsc@icloud.com (T.S.); takahiro-takazono@nagasaki-u.ac.jp (T.T.); koizumik@nagasaki-u.ac.jp (K.I.); 2Infection Control and Education Center, Nagasaki University Hospital, Nagasaki 852-8501, Japan; satoshi-kakiuchi@nagasaki-u.ac.jp (S.K.); ayumi.f@nagasaki-u.ac.jp (A.F.); ttakeshi@nagasaki-u.ac.jp (T.T.); 3Department of Respiratory Medicine, Nagasaki University Hospital, Nagasaki 852-8501, Japan

**Keywords:** azole-resistant *Aspergillus fumigatus*, environmental surveillance, hospital infection control, antifungal stewardship, PDCA cycle

## Abstract

As azole-resistant *Aspergillus fumigatus* emerges globally, healthcare facilities face mounting challenges in managing invasive aspergillosis. This review synthesizes worldwide azole resistance data to reveal profound regional variability, demonstrating that findings from other regions cannot be directly extrapolated to local settings. Consequently, hospital-level environmental surveillance is crucial for tailoring interventions to local epidemiology and detecting resistant strains in real-time. We outline practical approaches—encompassing sampling site prioritization, diagnostic workflows (culture-based and molecular), and PDCA-driven continuous improvement—so that even resource-limited facilities can manage resistant isolates more effectively. By linking real-time surveillance findings with clinical decisions, hospitals can tailor antifungal stewardship programs and swiftly adjust prophylaxis or treatment regimens. Our approach aims to enable accurate, ongoing evaluations of emerging resistance patterns, ensuring that institutions maintain efficient and adaptive programs. Ultimately, we advocate for sustained, collaborative efforts worldwide, where facilities adapt protocols to local conditions, share data through international networks, and contribute to a global knowledge base on resistance mechanisms. Through consistent application of these recommendations, healthcare systems can better preserve azole efficacy, safeguard immunocompromised populations, and refine infection control practices in the face of evolving challenges.

## 1. Introduction

### 1.1. Background and Rationale

The global healthcare community has witnessed an alarming escalation in the prevalence of azole-resistant *Aspergillus fumigatus* over the past decade [1]. *A. fumigatus*, a ubiquitous mold found in various environmental reservoirs, has long been recognized as a leading cause of invasive aspergillosis, particularly among immunocompromised and critically ill patient populations [2]. Importantly, *A. fumigatus* can form robust biofilms, a key virulence factor, particularly in invasive pulmonary aspergillosis and aspergilloma, which confers increased tolerance to antifungal agents and host immune defenses [3]. Historically, azole antifungal agents—such as itraconazole, voriconazole, posaconazole, and more recently isavuconazole—have formed the cornerstone of invasive aspergillosis prophylaxis and treatment [4,5,6]. However, the emergence and spread of azole-resistant strains now threaten to undermine this cornerstone, resulting in more frequent treatment failures and increased morbidity and mortality (Table 1, Key Message 1) [7,8]. In response to these global trends, several health agencies (e.g., the World Health Organization and Centers for Disease Control and Prevention) have highlighted azole-resistant *A. fumigatus* as a growing threat that requires urgent attention [9,10]. Nevertheless, practical guidance on environmental surveillance remains fragmented.

### 1.2. Clinical Implications and Healthcare Challenges

Several factors contribute to this growing challenge [11]. Intensive and prolonged use of azoles in clinical practice exerts strong selective pressure, driving the evolution of resistant fungal populations [12]. Concurrently, the widespread application of azole-based fungicides in agriculture creates environmental niches that favor the emergence of resistant *A. fumigatus* well before these strains enter healthcare facilities [13]. As a result, patients may inhale resistant conidia from their surroundings, leading to infections that no longer respond to standard antifungal therapy [14,15]. The interplay between environmental and clinical domains underscores the urgent need for more effective strategies to identify and control azole-resistant *A. fumigatus* within the hospital setting [16]. However, as new resistance mechanisms continue to emerge, there remains a gap in standardized approaches to environmental monitoring and prompt clinical intervention [17].

### 1.3. Gaps in Current Surveillance Strategies

Despite increasing recognition of azole resistance, systematic environmental surveillance for *A. fumigatus*, particularly aimed at detecting resistant strains, remains uncommon in most healthcare facilities [18]. In the absence of established routine monitoring protocols, standardized approaches to detecting and characterizing environmental fungal populations have yet to be widely adopted [19,20]. Moreover, the complexity of hospital environments—encompassing various potential reservoirs, from ventilation systems to construction areas—further complicates efforts to implement cost-effective and sustainable surveillance strategies. As healthcare institutions face limited resources and evolving clinical demands, there is a pressing need for a clear, practical roadmap that integrates evidence-based surveillance methods, rapid diagnostic techniques, and targeted environmental interventions.

### 1.4. Aim and Scope of This Review

While numerous studies have highlighted the clinical and public health impacts of azole resistance, few have offered a comprehensive, step-by-step framework for implementing routine environmental surveillance and early detection strategies within hospital settings. In response, this review addresses these gaps by providing a critical synthesis of existing evidence—including global prevalence data and diagnostic methodologies—and by translating these findings into a practical, operational model for hospital clinicians, infection control teams, and diagnostic laboratories. Specifically, we integrate cutting-edge detection methods, molecular epidemiological insights, and pragmatic implementation strategies to offer a multi-phase roadmap adaptable to varying healthcare settings, resource levels, and regulatory landscapes. Ultimately, by adopting these enhanced surveillance measures, institutions may reduce the incidence of resistant fungal infections, lower healthcare costs, and improve patient outcomes. In the sections that follow, we first discuss the clinical and public health significance of azole-resistant *A. fumigatus* and delve into its molecular mechanisms and epidemiology. We then propose optimized environmental surveillance strategies—including diagnostic methods and PDCA-based continuous quality improvement frameworks—followed by sections on future directions and a concluding summary. For an overview of the core challenges and practical recommendations regarding environmental surveillance for azole-resistant *A. fumigatus*, we have summarized key messages in Table 1.

## 2. Clinical and Public Health Significance

*A. fumigatus* poses significant challenges in both clinical and public health domains [9]. As highlighted in Key Message 1 (Table 1), azole-resistant strains of *A. fumigatus* represent a growing global threat, particularly for immunocompromised populations. Indeed, the World Health Organization (WHO) has underscored that “The emergence of resistance is partly driven by inappropriate antifungal use across the One Health spectrum. For example, agricultural use is responsible for rising rates of azole-resistant *Aspergillus fumigatus* infections, with azole-resistance rates of 15–20% reported in parts of Europe and over 80% in environmental samples in Asia” [9]. In parallel, both the European Centre for Disease Prevention and Control (ECDC) and the U.S. Centers for Disease Control and Prevention (CDC) have also emphasized the need for strengthened surveillance to address the spread of azole-resistant *A. fumigatus*, given its potential to undermine existing antifungal therapies and escalate clinical burden [10,21]. This collective stance highlights both the global scope of the problem and the multifaceted drivers of resistance.

### 2.1. Clinical Impact on Patient Outcomes

Clinically, the presence of azole-resistant strains has been linked to reduced antifungal efficacy, increased treatment failures, and worsened patient outcomes [8]. Patients with invasive aspergillosis often belong to high-risk groups such as those undergoing hematopoietic stem cell transplantation, solid organ transplantation, or receiving prolonged corticosteroid therapy [22]. Notably, among solid organ transplant recipients, lung transplant patients face particularly high risk for aspergillosis [23]. This heightened susceptibility stems from the lung’s direct and continuous exposure to inhaled environmental pathogens, compounded by the intensive immunosuppression required post-transplant [24]. Although data linking azole-resistant *A. fumigatus* and post-lung transplant infections remain limited [25], prophylactic antifungal management is recognized as pivotal in this population [5,26]. Early detection of environmental contamination and rapid intervention may thus be especially critical for lung transplant recipients. By highlighting the unique vulnerability of this group, hospitals and clinicians can tailor their surveillance protocols—such as more frequent environmental sampling or stricter airborne containment strategies—to better safeguard these patients. Additionally, ICU patients with severe influenza pneumonia or COVID-19 have emerged as another high-risk group for invasive aspergillosis, given the pulmonary damage and immunological dysregulation they often experience [27]. Mortality rates associated with invasive aspergillosis caused by azole-resistant *A. fumigatus* can increase significantly—some studies report rates exceeding 50% when first-line azole therapy fails—emphasizing the severity of this emerging threat [8,28,29]. In such cases, clinicians must resort to second-line or combination therapies that may be less effective, carry higher toxicity, or be more expensive [30]. These therapeutic limitations can lead to prolonged hospital stays, increased healthcare costs, and elevated morbidity and mortality. From a cost-effectiveness standpoint, undetected or late-detected azole resistance thus imposes a substantial financial burden on healthcare systems, further justifying the need for proactive surveillance.

### 2.2. Public Health Concerns and Environmental Origins

From a public health standpoint, the environmental origin of azole-resistant *A. fumigatus* underscores the complexity of controlling its spread. WHO has notably highlighted that inappropriate azole usage in agricultural settings contributes to expanding environmental reservoirs of resistant strains, which can then infiltrate healthcare facilities [9]. As emphasized in Key Message 2 (Table 1), factors such as agricultural fungicide use and imported plant bulbs can significantly contribute to the dissemination of resistant *A. fumigatus* strains [31,32]. These spores can travel long distances through the air, infiltrate hospital ventilation systems, and colonize construction areas, thereby posing a persistent threat to susceptible patients [33]. Once these resistant genotypes become established, they may spread further via spore-laden dust, complicating infection control efforts. Additionally, resistant strains can enter new regions through global trade networks, dispersing resistance traits across borders [32]. Such transboundary spread highlights the need for integrated, multinational strategies that align clinical surveillance, agricultural policies, and environmental interventions.

Importantly, a recent study reported that some azole-resistant *A. fumigatus* isolates exhibit no detectable fitness cost and may even demonstrate significantly increased competitive fitness in azole-free environments [34]. This finding challenges the conventional assumption that resistance mutations generally incur a biological trade-off. Should these high-fitness-resistant isolates become established within hospital or agricultural settings, the global spread of azole-resistant *A. fumigatus* could accelerate more rapidly than previously anticipated. Consequently, it is imperative to establish robust hospital-based environmental surveillance programs now—while the prevalence of these strains may still be relatively low—to mitigate the risk of large-scale dissemination and ensure more effective antifungal stewardship in the future.

### 2.3. Global Prevalence and Regional Variations

Azole resistance rates in *A. fumigatus* vary considerably by region, reflecting differences in agricultural practices, antifungal usage patterns, and the intensity of local surveillance efforts. Table 2 collates reported resistance rates from multiple countries and time periods but interpreting this data requires caution. Notably, the term “azole resistance rate” varies across studies: some define it differently or rely on methods other than EUCAST/CLSI microdilution. In order to compare data on a uniform basis, we extracted only results measured via EUCAST or CLSI protocols, using voriconazole (the first-line agent for invasive aspergillosis) as the reference drug. Specifically, we recalculated the percentage of isolates with MIC ≥ 2 µg/mL and MIC ≥ 4 µg/mL by reviewing each paper’s main text, tables, figures, or supplementary data. As a result, the rates in Table 2 may not match the original reported numbers exactly, but they follow a consistent standard that facilitates cross-regional and temporal comparisons.

**Africa**: Much of the existing data focus on environmental rather than clinical isolates, and standardized clinical surveillance remains insufficient. Consequently, the true impact of resistant *A. fumigatus* in patient care is not fully understood.**North America**: Most studies show relatively low clinical resistance rates, often below 5% [35,36,37,38]. However, resistant strains have occasionally been detected in environmental samples, indicating the need for vigilance to anticipate any upward trend.**South America**: Certain countries, especially Brazil, report high azole resistance among clinical isolates (exceeding 10% in some studies), underscoring the urgency of hospital-based monitoring in these high-incidence areas [39,40]. Other parts of South America have sparse data, making regional comparisons difficult.**Asia**: Japan and China stand out with notable rates of clinical resistance—sometimes above 10% or even 20% [41,42,43]. However, it is important to note that much of Japan’s azole resistance data comes from chronic pulmonary aspergillosis patients, who often receive long-term antifungal therapy; thus, these rates might be higher than those observed in invasive aspergillosis [41]. In fact, environmental isolates in Japan consistently show resistance rates below 10%, suggesting that the true rate of azole resistance in invasive aspergillosis could be lower than previously reported [44]. By contrast, environmental surveillance in other parts of Asia—particularly China, Taiwan, Thailand, and Vietnam—has documented resistance rates exceeding 30% [45,46,47], and in Vietnam, extremely high rates approaching 80% have been reported [48]. Such findings raise significant concern about the potential impact of resistant strains on clinical outcomes in these regions. Environmental surveillance has also revealed substantial pockets of resistant strains elsewhere in Asia, further underscoring the importance of a combined approach that integrates both clinical and environmental data.**Europe**: The Netherlands historically drew attention by documenting over 10% resistance among clinical isolates [17], and recent findings from Spain suggest a rising trend in certain hospitals [49]. Other European nations often report lower rates (<10%) but are not immune to localized increases [38], calling for continuous or prospective surveillance.**Middle East**: Reports from Iran suggest that, since 2016, both clinical and environmental *A. fumigatus* isolates appear to exhibit increasing azole resistance. In particular, some environmental samples have shown resistance rates exceeding 50% [50,51].**Oceania**: Available studies point to relatively low resistance, though data remain limited. Without expanded surveillance, shifts could go unrecognized.

Overall, Table 2 highlights that azole resistance is not a uniform phenomenon but rather a mosaic of local epidemiologies. Each region’s data reinforce the need for locally tailored surveillance strategies, cautioning against direct extrapolation of external rates to one’s own setting. Strengthening and harmonizing data collection, sampling methods, and reporting practices will facilitate more accurate temporal and cross-regional comparisons in the future.

**Table 2 jof-11-00096-t002:** Global prevalence and reported resistance rates of *Aspergillus fumigatus*.

Region	Pub. Year	Study Period	Cat.	No. Tested	Resistance Rate		PMID	Ref.
					VRC ≥ 2	VRC ≥ 4		
**Africa**								
Benin	2021	–	E	25	0.0%	0.0%	33921497	[52]
Burkina Faso	2024	2021–2022	E	124	3.2%	0.8%	38712846	[53]
Burkina Faso	2021	2019	E	646	0.2%	0.0%	33668719	[54]
Kenya	2018	–	E	48	14.6%	12.5%	30046310	[55]
Nigeria	2021	2017	E	5	0.0%	0.0%	34532039	[56]
Nigeria	2021	–	E	46	2.2%	0.0%	33921497	[52]
**North America**								
Canada	2020	2000–2013	C	985	0.1%	0.0%	31891387	[35]
Canada	2017	–	C + E	195	0.0%	0.0%	29156151	[57]
Mexico	2021	–	E	102	6.9%	5.9%	33921497	[52]
Mexico	2019	2014–2017	C	24	0.0%	0.0%	31220262	[36]
USA	2024	2019–2021	E	525	5.0%	4.2%	38651929	[58]
USA	2024	2018–2019	E	202	12.4%	–	38557086	[59]
USA	2024	2019	E	178	0.0%	0.0%	38534143	[60]
USA	2022	2015–2020	C	1891	3.3%	–	35400175	[37]
USA	2018	2015–2017	C	1356	0.1%	0.0%	29463545	[61]
USA	2015	2001–2014	C	220	8.2%	7.3%	26491179	[62]
2 countries	2024	2017–2021	C	282	1.4%	0.4%	38193696	[38]
**South America**								
Argentina	2020	2016–2019	C	93	8.6%	2.2%	32648614	[63]
Brazil	2024	2017–2019	C	27	0.0%	0.0%	39490213	[64]
Brazil	2023	2013–2019	C	40	37.5%	7.5%	37998875	[39]
Brazil	2023	–	C	84	1.2%	0.0%	36297597	[65]
Brazil	2020	2014–2017	C	199	1.0%	0.5%	31871090	[66]
Brazil	2018	1998–2014	C	168	23.8%	5.4%	29468746	[40]
Brazil	2017	–	C	25	4.0%	4.0%	29172033	[67]
Brazil	2017	–	E	20	0.0%	0.0%	29172033	[67]
Brazil	2017	1998–2017	C	221	1.8%	0.0%	28893772	[68]
Chile	2024	2017–2021	C	23	4.3%	4.3%	39304433	[69]
Paraguay	2021	–	E	36	8.3%	2.8%	33921497	[52]
Peru	2021	–	E	61	9.8%	8.2%	33921497	[52]
Peru	2019	–	C	143	0.7%	0.0%	31329931	[70]
**Asia**								
China	2024	2021–2023	C	276	0.7%	0.7%	38328338	[71]
China	2024	2021–2022	C	54	0.0%	0.0%	39637619	[72]
China	2024	2020–2023	C/E	94/251	18.8%	–	39470286	[73]
China	2024	2018–2022	C	146	13.7%	–	39669311	[43]
China	2023	2019–2021	C	81	6.2%	6.2%	37484905	[74]
China	2023	–	C	252	1.6%	1.6%	37580143	[75]
China	2023	2020	E	331	–	3.6%	37341484	[76]
China	2022	2019–2021	C	73	0.0%	0.0%	35493371	[77]
China	2021	1999–2019	C	445	2.0%	1.1%	34367087	[78]
China	2021	2019	E	233	38.6%	33.9%	33568450	[47]
China	2021	2018	E	1520	5.1%	4.4%	33544588	[79]
China	2020	2018	E	134	23.1%	9.0%	32718960	[80]
China	2020	2018	E	206	8.3%	7.3%	31855142	[81]
China	2020	2016	E	105	12.4%	9.5%	32718960	[80]
China	2020	2014	E	43	0.0%	0.0%	32718960	[80]
China	2017	2012–2015	C	126	2.4%	0.8%	28303848	[82]
China	2017	2011–2015	C	159	0.6%	0.0%	29209054	[83]
China	2016	2014–2015	E	144	0.7%	0.0%	27431231	[84]
India	2024	2018–2019	C	9	0.0%	0.0%	38524068	[85]
India	2024	2018–2019	E	3	0.0%	0.0%	38524068	[85]
India	2018	2012–2016	C	32	3.1%	3.1%	29891597	[86]
India	2015	2011–2014	C	685	1.6%	1.5%	26005442	[87]
India	2011	2005–2010	C	103	1.9%	0.0%	22028200	[88]
Indonesia	2021	2012–2015	C	8	0.0%	0.0%	34343127	[89]
Japan	2021	1996–2017	C	240	9.2%	2.5%	33309631	[90]
Japan	2020	2012–2019	C	120	21.7%	4.2%	32642756	[41]
Japan	2020	2017	E	203	6.9%	6.9%	32576436	[32]
Japan	2020	2013–2018	C	66	4.5%	3.0%	31564504	[91]
Japan	2019	2017–2018	C	55	23.6%	12.7%	30690480	[42]
Japan	2016	2013–2015	C	22	9.1%	4.5%	27050399	[44]
Japan	2016	2015	E	91	0.0%	0.0%	27050399	[44]
Japan	2014	1987–2008	C	171	0.6%	0.0%	24751235	[92]
Japan	2011	1994–2010	C	196	4.1%	0.0%	22024829	[93]
Korea	2020	2012–2013	C	84	0.0%	0.0%	32363043	[94]
Malaysia	2024	2019–2023	C	60	0.0%	0.0%	38828376	[95]
Taiwan	2023	2016–2020	C	118	4.2%	2.5%	37186489	[96]
Taiwan	2022	2015–2021	C	114	1.8%	0.9%	36135633	[97]
Taiwan	2019	2016–2018	C	14	0.0%	0.0%	31549427	[45]
Taiwan	2019	2016–2018	E	90	7.8%	5.6%	31549427	[45]
Taiwan	2015	2011–2014	C	38	7.9%	2.6%	26214171	[98]
Thailand	2023	2021	E	62	27.4%	3.2%	36254865	[46]
Thailand	2016	2014–2015	E	99	1.0%	0.0%	27664994	[99]
Vietnam	2021	2019	E	62	77.4%	59.7%	34232541	[48]
**Europe**								
Austria	2022	2020–2021	C	22	0.0%	0.0%	35205848	[100]
Belgium	2024	2020–2024	A	152	3.3%	2.0%	39430230	[101]
Belgium	2024	2020–2022	E	2937	0.2%	0.2%	38769604	[102]
Belgium	2023	2021–2022	A/E	35/68	10.8%	4.9%	36978451	[103]
Belgium	2021	2016–2020	C	1192	7.1%	–	34518094	[104]
Belgium	2017	2015–2016	C	109	10.1%	4.6%	28515220	[105]
Belgium	2015	2011–2012	C	192	6.8%	3.1%	25987612	[106]
Denmark	2024	2020–2022	E	4538	4.1%	–	38935978	[107]
Denmark	2022	2018–2020	C	1820	5.6%	3.9%	35104010	[108]
Danmark	2020	2018	C	137	8.8%	7.3%	32903400	[109]
Denmark	2020	2018	C	742	5.3%	3.0%	32556315	[110]
Denmark	2016	2007–2014	C	1098	3.6%	2.0%	27091095	[111]
Denmark	2016	2014	E	133	3.0%	3.0%	27091095	[111]
Denmark	2014	2010–2013	E	113	0.0%	0.0%	24936595	[112]
France	2023	2020–2021	E	166	0.0%	0.0%	37367554	[113]
France	2021	2015–2019	C	927	4.1%	3.0%	33680981	[114]
France	2021	2014–2018	C	35	8.6%	2.9%	33946598	[115]
France	2021	2014–2018	E	98	2.0%	1.0%	33946598	[115]
France	2021	2017	C	195	2.1%	2.1%	34619334	[116]
France	2019	2015	C	355	6.5%	5.4%	31038164	[117]
France	2018	2017	E	566	7.6%	4.9%	30215210	[118]
France	2018	2015	E	388	0.0%	0.0%	29853288	[119]
France	2018	2015	C	12	0.0%	0.0%	29853288	[119]
France	2017	2014–2016	E	157	14.0%	13.4%	28497646	[120]
France	2017	2011–2015	C + E	116	3.4%	3.4%	29082624	[121]
France	2015	2012	C	165	1.8%	1.8%	26026171	[122]
France	2011	2010–2011	C	131	3.1%	2.3%	22123701	[123]
France	2011	–	C	118	0.0%	0.0%	21131690	[124]
Germany	2020	–	A	159	0.6%	0.6%	32497229	[125]
Germany	2018	2012–2016	C	2888	3.2%	2.9%	29684150	[126]
Germany	2016	2011–2013	C	77	2.6%	1.3%	27989379	[127]
Germany	2015	2012–2013	C	27	29.6%	14.8%	25630644	[128]
Germany	2013	2011–2012	C	527	1.9%	0.9%	23669382	[129]
Greece	2020	2016–2017	E	101	1.0%	1.0%	32814940	[130]
Italy	2021	2016–2018	C	286	4.9%	1.4%	33438319	[131]
Italy	2020	2014–2016	C	134	0.0%	0.0%	32061880	[132]
Italy	2016	2013–2015	C	423	2.1%	0.5%	27356848	[133]
Italy	2016	1995–2006	C	533	3.4%	1.1%	26552980	[134]
Netherlands	2024	2019–2022	C	1850	15.7%	–	39644643	[135]
Netherlands	2024	2015–2020	A	142	11.3%	9.9%	38864903	[136]
Netherlands	2020	2018	C	764	17.8%	11.4%	32568033	[17]
Netherlands	2020	2017	C	774	19.1%	13.4%	32568033	[17]
Netherlands	2020	2016	C	784	16.1%	14.0%	32568033	[17]
Netherlands	2020	2015	C	600	11.8%	10.3%	32568033	[17]
Netherlands	2020	2014	C	814	8.6%	8.2%	32568033	[17]
Netherlands	2020	2013	C	760	9.3%	8.9%	32568033	[17]
Netherlands	2019	2011–2015	C	196	–	18.9%	30307492	[8]
Netherlands	2018	2006–2012	C	47	8.5%	4.3%	29394399	[137]
Netherlands	2018	2001–2017	C	363	39.1%	28.1%	30158470	[138]
Netherlands	2016	2010–2013	C	38	26.3%	26.3%	27541498	[139]
Netherlands	2015	2011–2013	C	105	22.9%	16.2%	26163402	[140]
Poland	2023	2015, 2019	E	31	0.0%	0.0%	37110454	[141]
Poland	2019	2015–2016	A	60	1.7%	1.7%	30616967	[142]
Poland	2017	2009–2011	C	121	2.5%	0.8%	28340159	[143]
Portugal	2021	2018–2019	E	99	3.0%	2.0%	33379247	[18]
Portugal	2021	2012–2019	C	70	2.9%	1.4%	33418997	[144]
Portugal	2021	2012–2019	E	39	10.3%	5.1%	33418997	[144]
Portugal	2019	–	E	31	0.0%	0.0%	31405297	[145]
Portugal	2019	–	E	55	0.0%	0.0%	30735287	[146]
Portugal	2018	2010–2016	C	190	3.2%	2.6%	30083151	[147]
Spain	2024	2023	C	3	66.7%	66.7%	38801514	[49]
Spain	2024	2019–2021	C/E	139/35	1.1%	1.1%	38551063	[148]
Spain	2021	2019	C	828	4.6%	0.7%	33010446	[149]
Spain	2019	2014–2018	C	158	7.0%	4.4%	31285229	[150]
Spain	2018	2017	C	260	–	0.8%	29941643	[151]
Spain	2013	1999–2011	C	362	5.0%	3.0%	23629706	[152]
Switzerland	2023	2019–2021	E	113	16.8%	12.4%	37930839	[153]
Switzerland	2022	2018–2019	C	355	1.1%	0.6%	35111868	[154]
Switzerland	2018	2016–2017	C	160	1.3%	1.3%	29437612	[155]
Turkey	2022	2018–2019	C	392	3.3%	3.1%	35445259	[156]
Turkey	2022	2018–2019	E	458	1.3%	1.3%	35445259	[156]
Turkey	2015	1999–2012	C	746	10.2%	–	26048062	[157]
UK	2023	2018–2019	E	2366	2.7%	0.5%	37478175	[158]
UK	2021	2018	E	146	0.0%	0.0%	33036151	[159]
UK	2018	2015–2017	C	356	1.1%	0.3%	30294314	[160]
UK	2018	2014–2016	C	167	6.0%	0.0%	30103005	[161]
UK	2018	2015	E	496	4.6%	1.4%	29997605	[162]
UK	2018	1998–2011	C	1151	0.3%	0.3%	30294314	[160]
3 countries	2024	2022–2023	E	2000	<10.0%	<10.0%	38842339	[163]
11 countries	2024	2017–2021	C	449	4.5%	1.1%	38193696	[38]
4 countries	2023	2020–2021	C	21	14.3%	9.5%	37998909	[164]
2 countries	2021	2015–2018	E	180	2.2%	2.2%	34835504	[165]
2 countries	2019	2012–2017	C	129	–	20.2%	31236587	[166]
**Middle East**								
Iran	2024	–	C	40	10.0%	2.5%	39239666	[167]
Iran	2024	2021–2022	E	37	51.4%	45.9%	38199436	[50]
Iran	2023	2021–2022	E	7	14.3%	14.3%	37713303	[168]
Iran	2023	2016–2021	C	23	0.0%	0.0%	36196507	[169]
Iran	2023	2016–2021	E	460	17.0%	–	36196507	[169]
Iran	2022	2018–2021	C	21	14.3%	0.0%	35579442	[170]
Iran	2021	–	E	60	80.0%	6.7%	33019714	[51]
Iran	2018	2009–2014	C	172	3.5%	2.9%	30181998	[171]
Iran	2016	2014	E	58	1.7%	0.0%	27656605	[172]
Iran	2016	2013–2015	C	71	2.8%	2.8%	27008655	[173]
Iran	2016	2013–2015	E	79	5.1%	5.1%	27008655	[173]
**Oceania**								
Australia	2024	2020–2023	C	169	5.3%	3.0%	39105545	[174]
Australia	2018	2015–2017	C	148	0.7%	0.7%	29846581	[175]
Australia	2018	2015–2017	A/E	11/41	0.0%	0.0%	29846581	[175]
New Zealand	2021	2001–2020	C	238	0.4%	0.4%	34140712	[176]
New Zealand	2021	2001–2019	C	210	1.4%	1.0%	33518383	[177]
2 countries	2023	2017–2020	C	46	8.7%	4.3%	37701716	[178]
**World**								
40 countries	2023	2017–2020	C	660	3.3%	0.9%	37367544	[179]
6 countries	2021	2011–2019	C	189	1.1%	–	34188199	[180]
29 countries	2017	2014–2015	C	391	0.3%	–	28784671	[181]
31 countries	2016	2013	C	142	1.4%	0.0%	27061369	[182]
9 countries	2015	2010–2012	C	6	16.7%	16.7%	25899126	[183]
France, China	2014	2010	E	175	0.0%	0.0%	24570417	[184]
39 countries	2011	2008–2009	C	497	1.8%	0.4%	21690285	[185]
>60 facilities	2010	2007–2009	C	607	1.6%	–	21123534	[186]
>60 facilities	2010	2004–2006	C	532	1.6%	–	21123534	[186]
>60 facilities	2010	2001–2003	C	173	0.0%	0.0%	21123534	[186]
3 countries	2010	–	C	2815	3.1%	1.4%	20592159	[187]
>60 facilities	2009	2005–2007	C	637	0.8%	–	19692559	[188]

Abbreviations: C = clinical isolate; E = environmental isolate; A = animal isolate; C + E = combined clinical and environmental isolates; Pub. Year = publication year; Cat. = category; No. Tested = number of isolates tested for susceptibility; VRC ≥ 2/VRC ≥ 4 (%) = proportion of isolates with voriconazole MIC ≥ 2 µg/mL or ≥4 µg/mL; PMID = PubMed ID; Ref. = reference. Methodology and Definitions: Only studies measuring VRC MIC by EUCAST microbroth dilution E.DEF 9.1 (or later) or CLSI M38-A2 (or later) were included. Resistance rates were calculated by the authors from each paper’s main text, tables, figures, and supplementary data, using the number of isolates tested for VRC susceptibility as the denominator. If *A. fumigatus* sensu stricto was distinguished, cryptic species were excluded; if not, isolates were recorded as the *A. fumigatus* complex. VRC, the first-line therapy for invasive aspergillosis, was chosen for chronological and geographic comparisons of azole resistance. Because “resistance” definitions of VRC vary over time, two MIC cutoffs (2 and 4 µg/mL) were adopted.

### 2.4. Implications for Stakeholders

Given these clinical and public health implications, stakeholders across healthcare, agriculture, and policymaking must recognize azole-resistant *A. fumigatus* as a pressing, multifaceted problem [9]. Clinicians need updated guidelines and rapid diagnostic tools for timely detection and appropriate therapy selection, while infection control teams require effective environmental surveillance strategies. In parallel, public health authorities and policymakers must address root causes by promoting judicious azole use in agriculture and supporting research into alternative fungicides or antifungal agents [21]. Only through coordinated, multidisciplinary efforts can the global burden of azole-resistant *A. fumigatus* be mitigated.

## 3. Mechanisms of Resistance and Molecular Epidemiology

### 3.1. Molecular Basis and Key Mutations

Azole resistance in *A. fumigatus* is primarily driven by genetic alterations in the cyp51A gene, which encodes the target enzyme of azole antifungals [189]. Frequently observed mutations, such as TR34/L98H and TR46/Y121F/T289A, involve tandem repeat (TR) insertions in the promoter region and point mutations in the coding sequence, respectively. These changes can increase cyp51A expression (via upregulation from the TR insertion) and/or alter the Cyp51A enzyme structure, thereby reducing azole binding affinity and conferring resistance [190]. TR-associated mutations are thought to arise under environmental selection pressure from azole fungicides, allowing resistant genotypes to proliferate before they reach clinical settings [191]. Notably, the presence of a tandem repeat in the promoter region is believed to result from sexual reproduction events in *A. fumigatus* [192], which occur only under specialized environmental conditions [193]. Because such reproductive conditions are generally absent in the human host, it is highly unlikely that TR mutations emerge de novo within the human body. Consequently, the detection of TR-related mutations (e.g., TR34 or TR46) strongly suggests that these isolates are of environmental origin rather than having evolved resistance in vivo [13]. Once selected in the environment, these resistant genotypes can be introduced into clinical domains via airborne conidia or contaminated plant materials, reinforcing the need for integrated surveillance that spans both settings [15]. Additionally, non-cyp51A-mediated mechanisms—including efflux pump overexpression and alterations in other sterol biosynthesis genes—may also contribute to resistance [189]. Recent reports highlight the potential role of mutations in genes such as hmg1, although their prevalence and clinical impact require further elucidation [194]. This diversity in molecular mechanisms complicates both diagnostic strategies and treatment approaches, underscoring the importance of continuous research into emerging or uncharacterized pathways.

### 3.2. Molecular Typing Methods and Their Utility

Tracing the origin and dissemination of resistant *A. fumigatus* strains requires robust molecular typing techniques. Historically, methods such as multilocus sequence typing (MLST)—which analyzes genetic variation at multiple housekeeping gene loci—have provided valuable insights into strain diversity and population structure [195]. However, MLST usage has declined in recent years for *A. fumigatus*, due in part to its lower discriminatory power and greater time requirement compared to microsatellite (short tandem repeat, STR) analysis [196]. STR-based analyses, which examine length variations in repeated DNA sequences, are now the most common epidemiological tool for *A. fumigatus*, offering a good balance of speed, cost, and resolution [197]. Meanwhile, whole-genome sequencing (WGS) provides a comprehensive overview of an organism’s genetic makeup, enabling the detection of subtle genetic differences and the identification of clonal lineages associated with resistance [15]. Although WGS offers higher resolution, it is also resource-intensive, making the choice of method dependent on local capabilities and specific epidemiological questions.

In addition to these broad typing approaches, targeted amplicon sequencing or rapid PCR-based genotyping can detect specific *cyp51A* mutations (e.g., TR34 or TR46) associated with azole resistance [198,199,200]. These assays, however, focus on known resistance-conferring variants rather than generating a strain-level epidemiological profile. As such, detecting TR34 or TR46 does not constitute a full “molecular typing” of the isolate but rather a targeted screening for key mutations. Ultimately, the balance between resolution, speed, and cost dictates which method laboratories adopt, with STR analysis remaining the mainstay for epidemiological typing and mutation-specific assays serving as rapid screens for resistance markers.

### 3.3. Linking Environmental and Clinical Isolates

A critical goal of molecular epidemiological studies is to establish a link between environmental and clinical strains, thereby clarifying infection sources and guiding prevention strategies. Comparative genomic analyses have demonstrated that certain resistant genotypes identified in environmental samples also appear in clinical isolates, strongly suggesting that patients can acquire infections from environmental reservoirs [15]. For example, clusters of genetically related resistant isolates have been detected in hospital environments, industrial sites such as sawmills, and agricultural products such as greenhouse-grown vegetables, which were subsequently matched to clinical cases of azole-resistant aspergillosis [73,201,202]. These findings not only highlight the ecological bridge between the environment and infected patients but also emphasize the importance of integrated surveillance programs that combine environmental sampling, molecular typing, and clinical case tracking. Regular application of these typing methods to environmental surveillance could identify emergent resistant clones before they become widespread in-patient populations, thus allowing targeted interventions at an earlier stage. By understanding the genetic relationships and pathways of strain dissemination, healthcare facilities can implement evidence-based strategies to reduce the introduction and spread of resistant *A. fumigatus*. Furthermore, a “One Health” perspective—which links human, animal, and environmental health—highlights how resistant strains can circulate across different domains [203,204,205]. This underscores the value of multisectoral collaboration in monitoring antifungal usage in agriculture, detecting early signs of emergent resistance in environmental niches, and swiftly translating these findings into clinical practice.

## 4. Optimizing Environmental Surveillance Strategies

### 4.1. Targeted Areas and Sampling Design

Establishing an effective environmental surveillance program for azole-resistant *A. fumigatus* requires careful consideration of where and how to sample. High-risk areas include intensive care units (ICUs) and hematopoietic stem cell transplant wards, where immunocompromised patients receive care and strict infection prevention measures are crucial [206,207]. Additional units with high-risk patient populations, such as solid organ transplant wards or those treating severe influenza/COVID-19 cases, may also warrant increased surveillance frequency. Ventilation systems, Heating, Ventilation, and Air Conditioning (HVAC) filters, and air handling units often serve as critical sampling points, as they can harbor fungal spores circulating throughout the hospital environment [208,209]. Construction and renovation sites within or adjacent to healthcare facilities are particularly important targets, as disturbances of building materials may mobilize fungal spores and increase exposure risk [210]. Developing a systematic sampling plan—including predefined locations, frequencies, and methodologies (e.g., air sampling, surface swabs, settle plates)—ensures consistent data collection and comparability over time [211]. In fact, the guidelines from the CDC in the United States recommend targeted sampling in areas with ongoing construction or renovation to preempt sudden increases in fungal burden [212]. By focusing on key hotspots and maintaining standardized protocols, hospitals can maximize resource efficiency while achieving reliable surveillance outcomes.

### 4.2. Addressing Air, Surfaces, Construction Activities, and Seasonal Variations

Environmental sampling strategies must also adapt to the complexities of the hospital microenvironment. Air sampling can detect airborne conidia released from HVAC systems or nearby construction zones, providing early warning of increased fungal loads [213]. Surface sampling—using swabs or contact plates—helps identify contamination reservoirs on medical equipment, building materials, and ventilation grilles [50,214]. During construction or renovation, temporary barriers, enhanced filtration, and targeted sampling at construction boundaries can mitigate the risk of azole-resistant *A. fumigatus* dissemination [212,215,216]. Seasonal factors, such as periods of high outdoor spore counts, increased agricultural activity [217], or monsoon seasons in tropical regions [218], may warrant more frequent sampling or heightened vigilance. By adjusting surveillance intensity in response to environmental changes, hospitals can proactively identify and address emerging threats.

Table 3 provides a comparative overview of the major sampling strategies for environmental surveillance—including air sampling (volumetric vs. settle plates), surface swabbing, contact plates, and adaptive approaches for construction or seasonal factors. This table outlines typical equipment requirements, advantages, limitations, and estimated cost/complexity associated with each method. Reviewing these options enables hospitals to select the most suitable combination of sampling techniques based on local resources, high-risk zones, and specific surveillance objectives.

### 4.3. Utilizing Checklists and Flowcharts for Surveillance Planning

The implementation of standardized checklists and flowcharts can streamline the surveillance planning process and promote consistent adherence to best practices [219]. These tools can outline key steps—from selecting sampling sites and determining sampling frequency to interpreting results and initiating interventions—ensuring that surveillance teams follow a clear, evidence-based protocol. Such standardization also facilitates staff training and helps newly assigned personnel rapidly understand and execute proper sampling procedures [220]. Flowcharts can help decision-makers rapidly assess whether additional sampling or remediation is needed based on real-time data, while checklists serve as reminders of essential tasks, such as calibrating equipment, documenting methodologies, and communicating results to relevant stakeholders [219]. By providing a structured, visual framework, these aids enable hospital teams to implement surveillance strategies more efficiently, respond to identified risks promptly, and continuously refine their approach as conditions evolve.

Table 4 presents a sample checklist detailing each phase of an environmental surveillance program—from defining objectives and assembling a multidisciplinary team to interpreting results and triggering interventions if necessary. Facilities can adapt or expand these steps based on local resources, risk profiles, and regulatory requirements. In conjunction with a decision-making flowchart, such as Figure 1, this checklist helps institutions maintain a systematic approach. By following a well-defined sequence of tasks and responsibilities, surveillance teams can better ensure data integrity, promptly address emergent threats, and foster continuous quality improvement through regular feedback loops. Figure 1 shows a concise decision-making flowchart for adjusting environmental surveillance based on routine sampling data, resistance thresholds, clinical spikes, and other contextual factors. Each yes/no decision leads to either maintaining current strategies or implementing additional interventions, ultimately converging on documentation and the Plan–Do–Check–Act (PDCA) cycle for continuous refinement.

This flowchart outlines a structured process for modifying hospital environmental surveillance in response to evolving data on azole-resistant *A. fumigatus*. Beginning with routine sampling results, each diamond poses a key yes/no question (e.g., “Any seasonal or construction risk?”, “Is the resistance rate above the threshold?”, “Any clinical spike or cluster?”), guiding teams toward either maintaining the current plan or implementing additional measures. All outcomes lead to documentation, communication, and an eventual return to the Plan–Do–Check–Act (PDCA) cycle for continuous quality improvement.

## 5. Diagnostic and Susceptibility Testing Methods

### 5.1. Comparison of Culture-Based and Non-Culture Methods

Accurate detection and characterization of azole-resistant *A. fumigatus* rely on appropriate diagnostic methodologies. Both culture-based and molecular diagnostics are indispensable in these settings, although culture remains the gold standard for detecting unknown or emerging mechanisms (Table 1, Key Message 3). Culture-based techniques—such as plating samples on selective media—allow for the recovery of viable fungi and subsequent susceptibility testing using standardized broth microdilution or agar-based assays [221]. These methods provide quantitative data on fungal load and enable direct antifungal susceptibility testing, but they can be time-consuming, often requiring several days for colony formation and subsequent analysis [222]. In contrast, non-culture approaches, such as PCR-based assays, offer rapid detection of resistance markers directly from samples without the need for fungal growth [223]. However, current evidence for these methods primarily derives from clinical specimens, and their direct application to environmental matrices remains underexplored. In addition, non-culture methods may lack quantitative precision regarding viable fungal load and can be more complex and costly, requiring specialized equipment and technical expertise [224]. Given these considerations, culture-based confirmation remains a critical component, particularly for identifying novel or uncharacterized resistance mechanisms that might evade narrowly targeted molecular tests [225]. Consequently, while non-culture diagnostics hold promise for accelerating detection, further validation is needed to establish their utility in routine environmental surveillance, where sample heterogeneity and lower fungal burdens pose additional challenges.

### 5.2. Microdilution Methods (EUCAST/CLSI Reference Procedures)

Microdilution assays for *A. fumigatus* are defined by procedures published by the European Committee on Antimicrobial Susceptibility Testing (EUCAST) and the Clinical and Laboratory Standards Institute (CLSI) [226,227]. Laboratories should select the protocols that best align with regional practices and available resources, ensuring consistent application to maintain data quality and comparability. EUCAST and CLSI protocols detail inoculum preparation, incubation conditions, and interpretive parameters required for reproducible susceptibility results. In practice, laboratories first determine the minimal inhibitory concentration (MIC) of each isolate via microdilution; the resulting MIC data are then compared against clinical breakpoints (BPs) or epidemiological cut-off values (ECVs) to classify the isolate’s susceptibility or non-wild-type status [228,229]. This quantitative MIC measurement is central to understanding azole resistance patterns in *A. fumigatus*. Furthermore, consistent adherence to quality control measures—such as using reference strains and validating assays with known resistant isolates—enhances the reliability of these results [230].

### 5.3. Clinical Breakpoints (BPs) and Epidemiological Cut-Off Values (ECVs)

BPs indicate whether an isolate is deemed susceptible, intermediate, or resistant, based on clinical outcome data and laboratory findings that inform therapeutic decisions [231]. They are periodically reviewed and updated as additional evidence on drug efficacy becomes available. Meanwhile, ECVs differentiate wild-type (WT) isolates from those likely harboring resistance-conferring mutations (non-WT) using population-based MIC distributions [232]. Unlike BPs, which focus on clinical response, ECVs simply highlight MIC values that deviate from the normal (WT) range for a given species. Table 5 outlines these BPs and ECVs for various azoles—including isavuconazole, itraconazole, posaconazole, and voriconazole—according to EUCAST and CLSI [228,229,233]. In environmental surveillance, ECVs may be especially beneficial when BPs are absent or less established for certain agents. By identifying isolates whose MIC values surpass the normal distribution range, laboratories can flag potentially resistant phenotypes, thereby prompting closer scrutiny or genetic analysis if needed.

### 5.4. Agar-Based Methods (Screening Plates and Related Approaches)

Agar-based methods can serve as an alternative or adjunct to microdilution. One common strategy involves screening plates containing fixed azole concentrations recommended by EUCAST or agencies such as the CDC [234,235]. Typical final concentrations include voriconazole 2 μg/mL, itraconazole 4 μg/mL, and posaconazole 0.5 μg/mL. After subculturing *A. fumigatus* onto these plates, colonies that grow under the selective pressure of azoles are flagged for subsequent confirmatory testing by microdilution. This two-step approach can detect resistant phenotypes relatively quickly but inherently involves an additional step before definitive MIC determinations. Some laboratories may also employ gradient diffusion strips (sometimes referred to as Etest or MIC test strips) to estimate MIC values on agar [236]. Regardless of the specific method chosen, careful subculture and inoculum standardization are essential. Where a full microdilution setup is readily available, it may be more time-efficient to proceed directly to microdilution rather than using agar screening first. Nonetheless, for labs with limited infrastructure, agar-based screening can be a practical gateway to identifying resistant isolates, especially if only a subset of suspicious colonies require further microdilution testing [237].

### 5.5. Introduction of Rapid and Sensitive Diagnostic Tools

While these standardized protocols and screening methods provide a solid framework for assessing antifungal susceptibility, the continually shifting epidemiology of azole-resistant *A. fumigatus*—combined with the demand for more rapid, sensitive diagnostics—underscores the importance of innovative tools in environmental contexts. As the urgency to detect azole-resistant *A. fumigatus* increases, recent developments in both culture-based and molecular diagnostic techniques offer new avenues to strengthen surveillance programs. The next section highlights these technological advancements and illustrates how they can help achieve more efficient and comprehensive monitoring. Ensuring that such new methods are adequately validated for environmental samples—and not just clinical isolates—will be crucial for large-scale adoption.

### 5.6. Culture-Based Improvements: Flamingo Medium

Zhang et al. (2021) developed Flamingo Medium; a selective agar designed to isolate *A. fumigatus* at elevated incubation temperatures (48 °C) while suppressing the growth of competing fungi such as Mucorales [238]. By incorporating compounds such as Rose Bengal and Dichloran, this medium effectively inhibits Mucorales and supports the rapid, robust development of *A. fumigatus* colonies. Studies have shown that Flamingo Medium recovers 20–30% more *A. fumigatus* colonies from environmental samples compared to conventional media, with visible colonies appearing within three days. Moreover, because Flamingo Medium is highly selective for *A. fumigatus*, laboratories can skip the microscopic identification process, thereby streamlining workflow and reducing the time needed to proceed with antifungal susceptibility testing. In environmental surveys, for example, any colony that grows on Flamingo Medium can be counted as *A. fumigatus*, and antifungal susceptibility testing of these isolates allows a straightforward calculation of overall resistance rates. This not only saves labor but also accelerates surveillance efforts, making Flamingo Medium an attractive option for laboratories seeking to enhance their detection and monitoring of *A. fumigatus* without the need for significant infrastructural or procedural modifications.

### 5.7. Molecular Assays: PCR, NGS, and Emerging Technologies

In parallel with culture-based improvements, molecular diagnostics have advanced to enable faster and more sensitive detection of azole-resistant *A. fumigatus*. For example, Gómez Londoño et al. (2023) developed nested PCR assays with high sensitivity and specificity for detecting TR-based resistance mutations (including TR34 and TR46 alleles) in the cyp51A gene directly from environmental samples—such as air, soil, compost, and plant debris [198]. These assays could detect as little as 5 fg of DNA, equivalent to a single fungal cell, and identified TR46 alleles in approximately 30% of tested samples, thus facilitating rapid environmental surveillance of azole-resistant *A. fumigatus* without the need for labor-intensive fungal isolation.

In addition to nested PCR, real-time PCR assays targeting cyp51A mutations have been validated primarily in clinical specimens and can produce actionable results within hours [200,239]. Next-generation sequencing (NGS) and targeted amplicon sequencing approaches offer even greater resolution, detecting low-abundance resistant variants and complex resistance patterns [240]. Additionally, emerging non-culture-based methods, such as loop-mediated isothermal amplification (LAMP) and CRISPR-based detection assays, are under investigation for their potential to deliver rapid, field-deployable diagnostics [241,242,243,244].

Another promising approach is matrix-assisted laser desorption/ionization time-of-flight mass spectrometry (MALDI-TOF MS). Recent work demonstrated that applying machine-learning techniques (e.g., partial least squares discriminant analysis) to MALDI-TOF protein spectra enabled not only the accurate discrimination of *A. fumigatus* sensu stricto from cryptic species but also the detection of azole-resistant isolates with high sensitivity and specificity [245]. While further validation is needed, this study underscores the potential of MALDI-TOF MS to provide rapid phenotypic screening for resistance, complementing the more targeted molecular assays described above.

By integrating improvements such as Flamingo Medium alongside advanced molecular techniques—ranging from nested PCR assays for direct TR detection in environmental samples to PCR, NGS, LAMP, MALDI-TOF MS, and CRISPR-based assays—routine surveillance programs can enhance early warning capabilities, guide more targeted containment measures, and ultimately improve patient outcomes. This tiered approach, starting with accessible culture enhancements and TR-targeted PCR for environmental matrices and progressing to high-resolution molecular diagnostics, provides a practical roadmap for hospitals and institutions to tailor their surveillance strategies based on available resources and evolving needs.

## 6. Data Interpretation and Integration into Clinical Practice

### 6.1. Setting Alert Thresholds and Risk Assessment Models

Translating environmental surveillance data into actionable strategies requires the establishment of well-defined alert thresholds and risk assessment models. These thresholds may be based on the frequency of azole-resistant *A. fumigatus* isolates recovered from specific sampling sites or on quantitative markers such as the concentration of conidia detected in air or surface samples [13]. For instance, an international expert panel has recommended that regions or facilities observing ≥10% azole resistance in *A. fumigatus* consider combination antifungal regimens or alternative therapies, reflecting a higher-risk scenario that warrants more aggressive interventions [30]. Additionally, Table 2 shows that in some countries, either environmental or clinical resistance rates have already reached or exceeded this 10% threshold. This observation suggests that adopting a similar benchmark for hospital-level decisions could be both practical and consistent with global data. Dynamic risk scoring systems can further adjust alert thresholds seasonally or in response to construction activities, local agricultural practices, or sudden spikes in resistance marker detection [73]. When these thresholds are reached or exceeded, they trigger predefined interventions, such as enhanced filtration, targeted environmental cleaning, or prophylactic antifungal therapy adjustments, thus facilitating a more proactive and cost-effective infection prevention strategy (Table 1, Key Message 4).

### 6.2. Multidisciplinary Decision-Making Processes

Effective integration of environmental surveillance data into clinical practice relies on collaborative decision-making among a range of stakeholders. Infection prevention and control teams, microbiology laboratories, infectious disease specialists, hospital administrators, and facility management staff must share and interpret data through regular multidisciplinary meetings or dedicated communication platforms [246]. Bringing together expertise from different fields ensures that results are not viewed in isolation but rather considered within the broader clinical and operational context. For instance, detecting a rise in azole-resistant strains in the ventilation system might prompt infection control teams to work closely with hospital engineers to inspect and upgrade air handling units, while clinicians may re-evaluate antifungal prophylaxis protocols for immunocompromised patients [247]. Such collaborative frameworks encourage timely and targeted responses to emerging threats, ultimately improving patient outcomes and resource allocation.

### 6.3. Practical Reporting Formats for Enhanced Clinical Responsiveness

Delivering surveillance findings to frontline clinicians and decision-makers in a clear, concise, and actionable format is crucial. Standardized reporting templates—incorporating data visualization tools such as trend graphs, heat maps, or color-coded alert levels—can help stakeholders quickly grasp the significance of new results [248]. For example, a monthly dashboard could summarize the frequency and distribution of azole-resistant isolates, highlight areas exceeding predefined alert thresholds, and provide suggested interventions based on established risk models. These reports can be disseminated through secure online portals or integrated directly into electronic health record systems, ensuring that relevant stakeholders receive timely updates. Providing interpretive guidance or concise action points—such as “Reassess first-line antifungal therapy for invasive aspergillosis in the Hematology Unit” or “Review prophylactic antifungal regimens in the Bone Marrow Transplant Department”—further enhances responsiveness [249]. By adopting standardized, user-friendly reporting formats, hospitals can facilitate rapid decision-making, encourage consistent application of best practices, and maintain an ongoing cycle of improvement in their environmental surveillance programs. Ultimately, these communication strategies ensure that critical resistance data inform not only clinical management but also long-term policy planning and resource allocation.

## 7. A Practical Framework and Recommended Strategies

### 7.1. PDCA Cycle for Continuous Quality Improvement of Environmental Surveillance

Implementing an environmental surveillance program for azole-resistant *A. fumigatus* requires not only a solid operational plan but also a mechanism to continually refine it in response to evolving data. The Plan–Do–Check–Act (PDCA) cycle offers a structured framework for ongoing quality improvement, ensuring that each surveillance cycle builds upon previous findings [250]. In Table 6, the four PDCA phases—Plan, Do, Check, and Act—are illustrated for an environmental surveillance context. Although PDCA operates continuously, not every cycle mandates substantial overhauls. Minor or major adjustments can be made depending on the magnitude of new findings, enabling the system to adapt proportionally to emerging challenges such as unexpected spikes in resistance or logistical constraints (Table 1, Key Message 5). By incorporating real-time surveillance data (e.g., from ongoing sampling or outbreak alerts), the PDCA cycle can be rapidly adjusted to mitigate potential risks before they escalate.

### 7.2. Action Flow for Clinical and Environmental Interventions

Whereas PDCA emphasizes improving the surveillance framework over repeated cycles, a separate but complementary pathway addresses real-time decisions prompted by threshold exceedances or clinical alerts. Figure 1 depicts a decision-making flowchart for promptly modifying surveillance and related interventions:**Routine Environmental Sampling**: Conducted on a scheduled basis (e.g., weekly or monthly).**Seasonal or Construction Risk?**: If present, intensify surveillance frequency or implement physical barriers and dust control [212].**Is the Resistance Rate Above the Threshold?**: If yes, consider not only prophylactic measures (e.g., adjusting antifungal prophylaxis for high-risk wards) but also revisiting the first-line therapy recommendations for invasive aspergillosis, especially in wards with high-risk patients [30].**Any Clinical Spike or Cluster?**: If a cluster is detected, “Enhanced Measures” may include thorough environmental cleaning, access restrictions, or further diagnostic evaluations [210].**Any Hotspot from Environmental Data?**: If local contamination is suspected, perform targeted remediation and repeat tests to ensure successful mitigation [251].**Document and Communicate**: Results are shared with relevant stakeholders (infection control teams, hospital administration), and the outcomes feed back into the PDCA cycle for subsequent refinement.

This integrated approach ensures that newly identified risks—whether environmental or clinical—are promptly addressed through clearly defined interventions and follow-up actions.

### 7.3. Example of a Two-Person Surveillance Plan

The following plan illustrates a practical example from our institution’s experience and may be adapted according to each hospital’s local environment and resources.

### 7.4. Selecting Surveillance Locations Under Limited Staffing

In our hospital, we chose to focus on two wards with distinct characteristics, recognizing that other facilities may have different configurations:**Hematology Ward (with HEPA Filtration)**Patients here often undergo hematopoietic stem cell transplantation and are therefore at high risk for invasive aspergillosis. However, this ward is equipped with HEPA filters, potentially resulting in very low yields of airborne *Aspergillus*.**Respiratory Ward (without HEPA Filtration)**This ward admits lung transplant recipients, who are also at high risk due to continuous exposure to inhaled pathogens and intensive immunosuppression. Unlike the hematology ward, the respiratory ward in our facility is not equipped with HEPA filters, making it easier to detect airborne fungi if present.

By selecting one ward with advanced filtration and another without it, we can capture a more comprehensive risk profile in our hospital. However, institutions with different filtration setups or patient distributions should adapt these choices to suit their local environment.

### 7.5. Workflow Overview

Figure 2 outlines how environmental sampling is conducted by a two-person team—one infection control specialist and one microbiology technician. The workflow includes:Regular air sampling (weekly)Triggered surface swabs (when an increase in *A. fumigatus* is detected)Soil sampling (if azole resistance exceeds 10%)

**Figure 2 jof-11-00096-f002:**
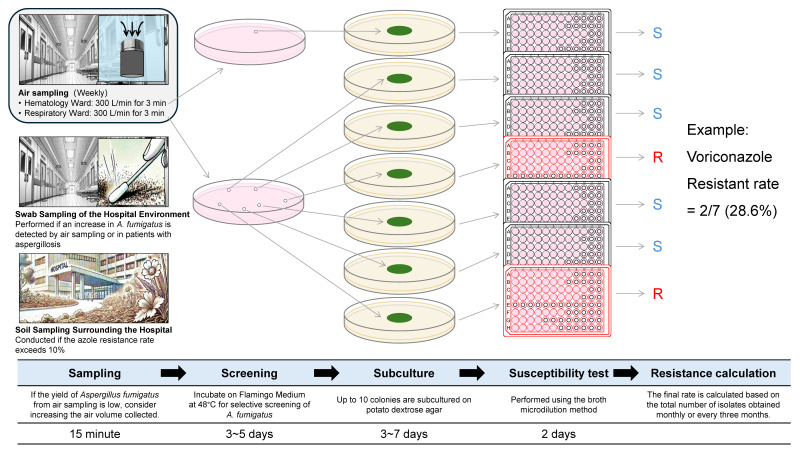
An example of an environmental surveillance workflow for azole-resistant *Aspergillus fumigatus* in a hospital.

Cultures are initially screened on Flamingo Medium (48 °C, 3–5 days), then subcultured on potato dextrose agar (3–7 days) [238]. A broth microdilution test (2 days) confirms azole resistance. Data from these procedures are integrated into the PDCA cycle, and immediate interventions are triggered if thresholds or clinical findings warrant a more aggressive response. Despite minimal staffing, this streamlined strategy demonstrates how routine surveillance can be maintained effectively—even in smaller or resource-limited facilities.

This figure illustrates one specific example of an environmental surveillance approach to monitor azole-resistant *A. fumigatus* in a hospital setting. Air sampling is conducted weekly in both the Hematology Ward and the Respiratory Ward, while swab sampling of the hospital environment is triggered if an increase in *A. fumigatus* is detected by air sampling or in patients with aspergillosis. Soil sampling is performed around the hospital if the azole resistance rate exceeds 10%. Samples are incubated on Flamingo Medium at 48 °C for 3–5 days to screen for *A. fumigatus* sensu stricto, and then up to ten colonies are subcultured on potato dextrose agar (3–7 days). Susceptibility testing follows, using the broth microdilution method (2 days). Blue-labeled “S” indicates susceptible isolates, and red-labeled “R” indicates resistant isolates. The azole resistance rate is finally calculated based on the proportion of resistant isolates among those tested.

### 7.6. Additional Soil-Sampling Protocol

In settings where soil is sampled from the hospital vicinity—especially if overall fungal loads are high—it is practical to use itraconazole and voriconazole selective media (e.g., Flamingo Medium supplemented with each agent) to reduce the workload in susceptibility testing. As shown in Figure 3, soil suspensions are spread on three plates: a control plate (Flamingo Medium only) and two plates containing itraconazole (4 μg/mL) or voriconazole (2 μg/mL). Only colonies that grow on these drug-supplemented plates are subsequently subcultured and undergo broth microdilution testing. This approach filters out drug-susceptible strains and focuses laboratory resources on isolates likely to harbor resistance, significantly cutting down on the volume of subcultures needed. In practice, 5 g of soil is mixed with 10 mL of saline containing 0.05% Tween 80, vortexed, and 50 µL of the supernatant is inoculated onto each selective plate [238]. Incubation on Flamingo Medium at 48 °C permits efficient detection of *A. fumigatus*, and any colonies growing on selective agar indicate potential itraconazole or voriconazole resistance [163]. This streamlined protocol aligns with the two-person workflow described above, supporting targeted, high-yield surveillance while keeping laboratory burdens manageable—even for facilities with limited personnel.

A schematic of one possible approach for screening *Aspergillus fumigatus* from hospital soil samples by employing Flamingo Medium supplemented with itraconazole (ITC, 4 µg/mL) or voriconazole (VRC, 2 µg/mL). Five grams of soil is mixed with 10 mL saline containing 0.05% Tween 80, vortexed, and 50 µL of the supernatant is plated onto three separate plates: one control plate (Flamingo Medium only) and two selective plates (Flamingo Medium + ITC or +VRC). The total colony count on the control plate serves as the denominator for calculating resistance rates, while only colonies growing on the drug-supplemented plates are subcultured for subsequent susceptibility testing. Red-labeled “R” indicates resistant isolates. This strategy focuses testing resources on isolates most likely to harbor azole resistance, thereby reducing overall workload. Note that this protocol serves as an illustrative example; hospitals may adapt or modify it according to local infrastructure and epidemiological contexts.

### 7.7. Linking Existing Resources and Adding Value

Internationally recognized protocols (e.g., EUCAST, CLSI) guide antifungal susceptibility testing but do not comprehensively address how environmental data integrate into broader hospital strategies [226,227]. The two-tiered approach—(1) a PDCA cycle for long-term surveillance refinement (Table 6) and (2) a decision-flow mechanism (Figure 1) for clinical or environmental interventions—bridges this gap by ensuring both ongoing quality improvement and swift, data-driven actions. Institutions that already maintain robust links between environmental findings and clinical decisions may find portions of this section to overlap with existing practices [213]. However, for settings developing or expanding such frameworks, the methods described serve as a blueprint for aligning resources, infrastructures, and response protocols to evolving resistance patterns. By leveraging existing microbiology expertise, infection control networks, and hospital management systems, facilities can adopt this two-tiered framework without reinventing entire procedures. As a result, both small community hospitals and large academic centers can benefit from consistent updates in environmental data, ensuring that clinical protocols remain agile and reflective of the local resistance landscape.

## 8. Cost-Effectiveness and Sustainability

### 8.1. Cost Evaluation Models

Implementing an environmental surveillance program for azole-resistant *A. fumigatus* inevitably requires financial and human resource commitments. Although direct cost analyses specific to azole-resistant strains remain limited, higher mortality rates associated with resistance suggest that missed or delayed detection can lead to increased resource utilization—such as extended hospital stays or additional antifungal therapies [8]. From an economic standpoint, investing in routine surveillance may help avert potential downstream costs and improve patient outcomes, even under budget constraints. As each institution’s budget and clinical priorities vary, tailored cost models can be developed to estimate both the expenses of maintaining the program (e.g., staff, equipment, testing) and the potential savings from early detection and intervention. This broader view of costs and benefits supports a strategic approach to resource allocation.

### 8.2. Resource Optimization Strategies

Given that many healthcare facilities operate under limited budgets and staff availability, optimizing existing resources is critical for long-term program viability. Several strategies may be considered:**Targeted Sampling Approaches**Focusing surveillance on high-risk wards (e.g., hematology or transplant units) or periods of known risk elevation (e.g., construction phases, seasonal variations) can reduce the need for extensive sampling across the entire hospital.**Leverage Multipurpose Equipment**Investing in a versatile air sampler or a molecular testing platform that can serve multiple surveillance or diagnostic purposes may be more cost-effective than acquiring single-use devices.**Task-Sharing**Training existing staff (e.g., infection control nurses, lab technicians) to handle basic sampling or preliminary lab work can distribute workload and minimize reliance on additional hires. Cross-training personnel also enhances institutional resilience by ensuring coverage during staff turnover or absences.**Collaborative Networks**Forming partnerships with regional laboratories or research institutions may allow for shared costs in molecular testing or advanced data analytics, thus reducing the financial burden on a single facility. Such networks can also facilitate benchmarking and data-sharing, potentially leading to multi-institutional studies that further justify surveillance investments.

These resource optimization tactics not only reduce immediate costs but also position the surveillance program to adapt more flexibly to new challenges, such as emerging resistance patterns or administrative changes.

### 8.3. Implications for Long-Term Sustainability

Sustaining a meaningful surveillance program over multiple years requires a strategic balance between cost containment and data quality. Key considerations include:**Budgeting for Expansion**As data accumulate and PDCA cycles reveal improvement opportunities, the program may need to scale up sampling or incorporate more advanced molecular techniques. Securing long-term funding or establishing contingency budgets is essential.**Continuous Training and Staff Retention**Knowledgeable personnel are pivotal to a successful surveillance program. Regular training, cross-disciplinary workshops, and career development pathways can improve staff motivation and reduce turnover. A stable, well-trained team also enhances data consistency over time.**Policy and Regulatory Support**National or regional guidelines that encourage routine fungal surveillance—and possibly allocate funding—can significantly bolster the feasibility of maintaining such programs. In some regions, compliance with mandatory reporting or accreditation standards may drive institutions to adopt systematic surveillance, securing financial backing along the way [252].**Public Health Impact**Reductions in azole-resistant *A. fumigatus* infections and related complications may produce intangible benefits, such as safeguarding institutional reputation and reinforcing public confidence in healthcare services. Moreover, the broader One Health implications—including agricultural and environmental considerations—can garner support from external stakeholders and policymakers.

By embedding cost-effectiveness and sustainability considerations into each PDCA cycle, hospitals can ensure that environmental surveillance remains a strategically valuable, evidence-based component of their infection prevention efforts. Ultimately, the balance between resource investment and clinical benefits will drive the long-term success of azole-resistance monitoring, as these considerations can be periodically revisited to keep the program both financially sustainable and clinically impactful.

## 9. Future Directions

### 9.1. Advancing Molecular Diagnostics While Preserving Culture-Based Methods

The emergence of azole-resistant *A. fumigatus* strains—most notably those involving TR34/L98H and TR46/Y121F/T289A—has underscored the potential value of rapid molecular diagnostics in hospital settings [199]. Although conventional culture and antifungal susceptibility testing remain time-consuming, they still serve as the gold standard, especially when novel or uncharacterized resistance mechanisms arise in the environment. While targeted molecular assays (e.g., isothermal amplification, next-generation PCR) can shorten turnaround times for detecting specific resistance mutations, they may not readily capture new or unexpected resistance pathways. Hence, these methods—once fully validated—could function as a complement to, rather than a replacement for, culture-based surveillance. Retaining culture-based assays ensures a safety net for emergent or cryptic resistance mechanisms that might not be covered by current molecular targets.

### 9.2. Strengthening Routine Surveillance in High-Risk Wards

In units where immunocompromised patients are concentrated, regular air or surface sampling remains essential to detect sudden increases in resistant *A. fumigatus* [18]. Incorporating emerging molecular diagnostics into these monitoring protocols may yield faster alerts for TR34- or TR46-based resistance. However, because unknown cyp51A mutations or other mechanisms might not be detected by narrowly targeted assays, culture-based methods provide a comprehensive safety net [16]. Periodic reviews of resource allocation, test frequency, and data quality can help ensure that the surveillance program remains both effective and sustainable. In some high-risk units, combining conventional culture with rapid molecular screens may offer the best balance of sensitivity, speed, and cost-effectiveness.

### 9.3. Real-World Validation and Technological Limitations

Although molecular diagnostics have shown promise, they are still largely in a research phase, particularly for direct environmental samples that typically contain low fungal loads or inhibitory substances [198]. Even if certain assays prove highly specific for TR34 or TR46, it remains uncertain whether they can reliably detect a broader range of future resistance mechanisms. Rigorous field validation and cost-effectiveness studies are required before large-scale adoption [253]. Consequently, culture-based methods—capable of capturing novel or unexpected phenotypes—will likely maintain a central role in hospital surveillance. To close these gaps, large multicenter studies focusing on real-world performance of molecular assays in diverse environmental matrices are critically needed.

### 9.4. International Collaboration and Rapid Adaptation

Azole-resistant *A. fumigatus* is shaped by international dynamics, notably agricultural fungicide usage and the import of agricultural goods carrying resistant strains, highlighting a possible route for intercontinental spread [32,191]. International data-sharing networks can significantly enhance early detection of newly emerging or increasing resistance mechanisms, not just TR34 or TR46, but also uncharacterized mutations that may arise in the future. By participating in regional or global platforms, healthcare institutions can promptly recognize shifts in resistance patterns and coordinate responses. If participating centers standardize molecular protocols (for instance, by agreeing on shared primer sets or using the same reference strains), they can compare results more accurately and issue timely alerts when novel cyp51A mutations or other pathways appear in multiple regions. This unified approach enables earlier detection and concerted action against newly circulating resistance traits (Table 1, Key Message 6). Moreover, broader One Health initiatives that link environmental, agricultural, and clinical data can further strengthen this collaborative surveillance infrastructure, fostering rapid adaptation to evolving resistance landscapes.

### 9.5. Ensuring Future-Ready Antifungal Stewardship

As new antifungals and combination therapies are introduced, the ability to rapidly identify resistant isolates—whether through improved culture-based screening or validated molecular assays—will become increasingly critical [30]. Local stewardship programs, guided by robust surveillance data, can optimize drug usage and mitigate the impact of resistant *A. fumigatus*. However, given the uncertain outlook for environmental azole usage, hospitals must remain vigilant—continuously refining their surveillance strategies and adopting new diagnostic tools as they become validated. By doing so, they can prepare for both recognized mutations (e.g., TR34, TR46) and other, uncharacterized resistance mechanisms yet to arise. Ultimately, a future-ready stewardship framework must integrate routine analysis of emerging resistance data, ensuring that clinicians are equipped with updated therapeutic options and best-practice guidelines for each newly recognized resistant phenotype.

## 10. Conclusions

Azole-resistant *A. fumigatus* has become a critical global concern, threatening patient outcomes by eroding the effectiveness of standard antifungal therapies. One of the major contributions of this review is its comprehensive synthesis of worldwide resistance rates (Table 2), which reveals significant regional variability—ranging from under 1% in some areas to nearly 80% in others. This disparity makes it clear that extrapolating data from other regions to one’s own facility is neither accurate nor sufficient. Consequently, the need for sustained, hospital-level environmental surveillance becomes all the more pressing.

Alongside these global prevalence insights, this review outlines practical strategies for detecting and managing resistant isolates at the institutional level. While molecular diagnostics can accelerate the identification of known mutations, culture-based methods remain crucial for uncovering novel or uncharacterized resistance mechanisms. Equally vital is long-term, locally tailored surveillance, periodically refined by PDCA to maintain cost-effectiveness and responsiveness to emerging threats. By integrating surveillance findings into clinical workflows—through actionable reporting formats, multidisciplinary decision-making, and antimicrobial stewardship—hospitals can better protect their most vulnerable patient populations.

Ultimately, azole resistance transcends institutional and geographic boundaries, and data-sharing networks as well as international collaboration are essential for early detection of newly circulating resistance traits. However, as this review demonstrates, the wide variability in regional resistance rates underscores that each hospital must implement its own robust environmental surveillance program rather than relying on external data alone. Through ongoing commitment to these local strategies, supported by global knowledge exchange, healthcare systems will be better equipped to preserve azole efficacy and safeguard patient outcomes in an ever-evolving landscape of antifungal resistance.

## Figures and Tables

**Figure 1 jof-11-00096-f001:**
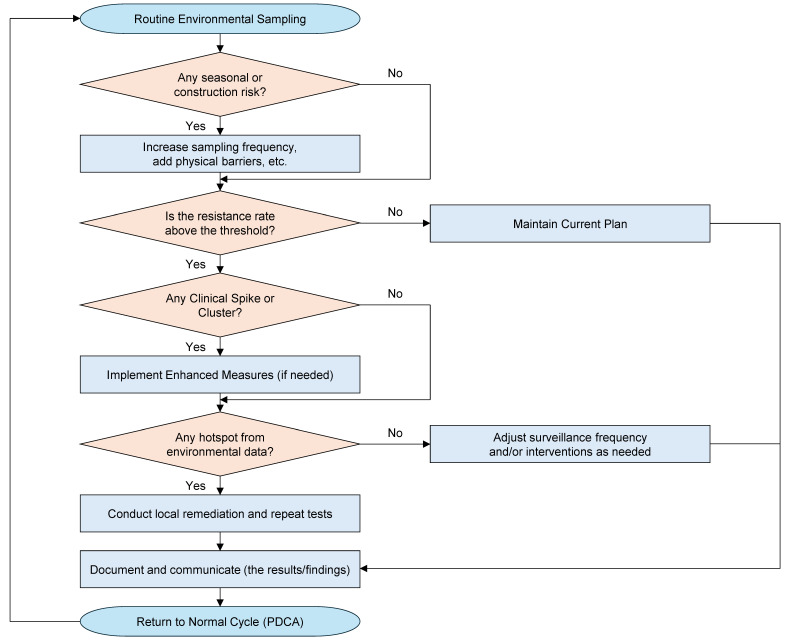
Decision-Making Flowchart for Adjusting Environmental Surveillance of Azole-Resistant *Aspergillus fumigatus*.

**Figure 3 jof-11-00096-f003:**
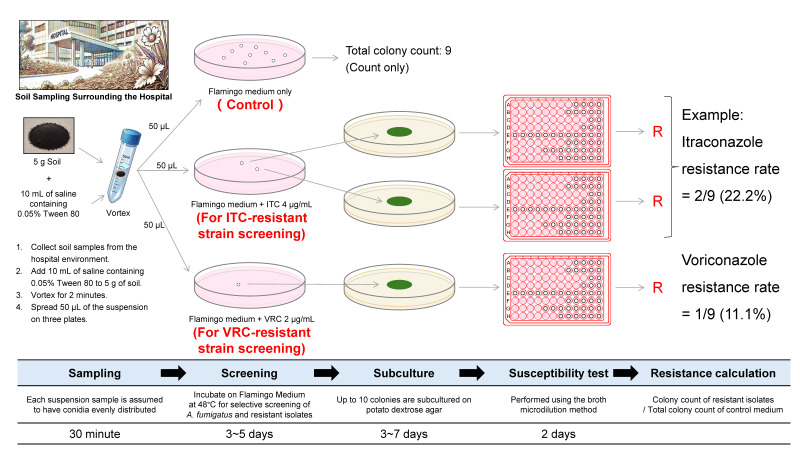
An example of soil sampling protocol using Flamingo Medium supplemented with ITC or VRC.

**Table 1 jof-11-00096-t001:** Key messages for environmental surveillance of azole-resistant *Aspergillus fumigatus*.

	Key Message	Implication for Practice
1	Azole-resistant *A. fumigatus* is a growing threat globally, especially for immunocompromised patients.	Hospitals should prioritize proactive surveillance, focusing on high-risk wards to prevent major outbreaks.
2	Environmental factors (agricultural fungicides, imported plant bulbs) contribute to resistant strain dissemination.	Collaboration with agricultural and public health authorities is needed to mitigate external sources of resistant spores.
3	Both culture-based and molecular diagnostics are crucial, but culture remains the gold standard.	While molecular methods speed up detection, they may not capture unknown mechanisms, so culture-based surveillance is indispensable.
4	Rapid identification of resistant strains is crucial for timely interventions and better patient outcomes.	Integrating surveillance results into clinical decisions enables timely drug regimen adjustments and reduces treatment failures.
5	Periodic reevaluation (e.g., using PDCA) ensures environmental surveillance stays cost-effective, sustainable, and responsive to new threats.	Surveillance programs must adapt sampling frequency, test methods, and interventions as local conditions change.
6	International and One Health–based data-sharing networks enhance early detection of emerging resistance and foster coordinated responses.	Standardizing protocols across centers allows rapid alerts and coordinated actions against newly identified resistance mechanisms or strains.

Abbreviations: PDCA = Plan-Do-Check-Act cycle.

**Table 3 jof-11-00096-t003:** Summary of sampling strategies for environmental surveillance of azole-resistant *Aspergillus fumigatus*.

Strategy/Method	Equipment and Approach	Advantages	Limitations and Cost
**Air sampling (volumetric)**	- Volumetric sampler (e.g., Andersen) drawing a controlled volume of air - Measures fungal load in CFU/m^3^ (CFU = Colony Forming Units)	- Quantitative data (low-level detection) - Ideal for long-term monitoring of airborne loads	- Equipment can cost USD 1000–10,000+ depending on model/maintenance - Requires calibration and trained staff - Labor-intensive (culture + colony counts) - Overall moderate to high cost
**Air sampling (settle plates)**	- Open agar plates left in the environment for passive spore settling (e.g., 30–60 min)	- Low cost and minimal setup - Straightforward deployment and interpretation - Early indicator of airborne “hot spots”	- Semi-quantitative only (approximate colony counts, not exact CFU/m^3^), so not precise for large-scale quantification - Dependent on airflow, sampling duration, and plate placement - Generally low cost and minimal training required
**Surface sampling (swab)**	- Sterile swabs on targeted surfaces (e.g., vents, shelves, corners) to detect localized fungal contamination	- Pinpoints localized “hot spots” - Fast, easy collection - Requires minimal equipment	- Primarily qualitative (presence/absence) - Accuracy varies with technique and surface area - Risk of cross-contamination if protocols are not followed - Low material cost, but lab/on-site culture needed for confirmation
**Surface sampling (contact plates)**	- Contact agar plates (e.g., Rodac) pressed onto flat surfaces	- Semi-quantitative detection in specific zones - Useful for routine checks in high-risk wards - Easy to standardize (pressure + area of contact)	- Requires smooth, flat surfaces (not suitable for porous or uneven materials) - Incubation + colony enumeration needed - Low to moderate cost (primarily plate costs + basic sterile technique)
**Seasonal/construction factors (adaptive strategy)**	- Adaptive scheduling (frequency, location) responding to seasonal spore peaks or construction phases (e.g., high humidity, dust)	- Enhanced detection during peak spore seasons (e.g., monsoon season) - Risk-based targeting of high-risk periods - Can align with infection control measures	- More complex scheduling/logistics - Resource-intensive if sampling frequency rises - Possible operational impact (area closures, extra filtration) - Cost varies widely, depending on site conditions and control measures

**Table 4 jof-11-00096-t004:** Sample checklist for environmental surveillance planning.

Step/Item	Key Actions/Considerations	Timing/Frequency	Notes
**1. Define Objectives and Scope**	- Identify critical wards or equipment (e.g., ICU, transplant wards, ventilation systems) - Establish performance goals (e.g., detect >10% azole-resistant isolates) - Align scope with broader infection control policies	Initial Phase/Start-up Phase	- Collaborate with relevant stakeholders (e.g., hospital management, facility maintenance) - Clarify specific targets (e.g., outbreak prevention)
**2. Assemble Surveillance Team**	- Assign key roles (sampling technician, data analyst, etc.) - Provide training on sampling protocol - Ensure multidisciplinary involvement (Infection Control Service, microbiology, facility/maintenance, etc.)	Initial Phase/Start-up Phase	- Allocate resources (budget, staff time) for effective training
**3** **. Select Sampling Methods**	- Choose air sampling (volumetric or settle plates) - Decide on surface sampling (swabs or contact plates) - Plan adaptive strategy for seasonal/construction factors	After scope defined	- Refer to Table 3 for method comparison (cost, complexity, desired data detail)
**4. Identify Sites and Frequency**	- Map high-risk zones (HVAC filters, construction areas, etc.) - Determine sampling intervals (weekly, monthly, event-based) - Conduct preliminary risk assessment (optional)	Before first sampling	- Create a site map indicating hotspots - Plan weekly, monthly, or event-triggered schedules
**5. Prepare Resources and Equipment**	- Calibrate air samplers (if applicable) - Stock agar plates, swabs, transport media, PPE - Confirm capacity for incoming samples	Just prior to sampling (e.g., the day before)	- Check expiry dates of agar plates, transport media - Ensure staff have correct PPE
**6. Conduct Sampling**	- Perform sampling at designated sites/times - Document any anomalies (e.g., construction dust, spills) - Staff must wear appropriate PPE	As per set frequency (e.g., weekly)	- Record date, time, location, conditions (temperature, humidity, etc.)
**7. Transport and Process Samples**	- Transport samples to the lab under correct conditions - Inoculate/culture for *Aspergillus fumigatus* detection	Same day as sampling	- Maintain chain of custody (i.e., documented traceability of samples) to ensure proper labeling, sealing, and tracking from collection to analysis. - Minimize transit time to lab
**8. Analyze Results and Compare with Thresholds**	- Cultivate and identify *A. fumigatus* sensu stricto using selective media (e.g., Flamingo Medium at 48 °C) - Once confirmed, count total colonies (CFU) for presence/absence - Conduct antifungal susceptibility testing (e.g., EUCAST/CLSI) or use azole-containing agar - Determine % of resistant isolates among total *A. fumigatus*	Post-incubation (72 h or as needed), plus additional time for azole susceptibility screening	- Follow official documents (EUCAST/CLSI) for susceptibility - Record growth data; interpret using established breakpoints or epidemiological cut-off values
**9. Interpret and Report Findings**	- Prepare summary (tables, graphs) - Highlight any results exceeding thresholds or unusual spikes	Within 1 week after lab results were finalized	- Provide clear, concise format for the Infection Control Committee or stakeholders
**10. Trigger Interventions if Needed**	- If thresholds exceeded (e.g., >10% resistant isolates), consider enhanced cleaning, filter changes, and restricted area access - Intensify surveillance or reevaluate sampling schedule	Immediately upon detection	- Additional measures (isolation, prophylaxis) may be required in outbreak scenarios
**11. Feedback and Continuous Improvement**	- Implement the PDCA cycle: review sampling efficiency, cost, and outcomes - Adjust plan if new hotspots or building works arise	Monthly or quarterly review	- Modify sampling protocols or frequency as needed based on results or facility changes

Abbreviations: ICU = Intensive Care Unit; HVAC = Heating, Ventilation, and Air Conditioning; EUCAST = European Committee on Antimicrobial Susceptibility Testing; CLSI = Clinical and Laboratory Standards Institute; CDC = Centers for Disease Control and Prevention; PPE = Personal Protective Equipment; PDCA = Plan–Do–Check–Act cycle for continuous quality improvement.

**Table 5 jof-11-00096-t005:** Clinical breakpoints (BPs) and epidemiological cutoff values (ECVs) against *Aspergillus fumigatus*.

	Clinical Breakpoints (μg/mL)						Epidemiological Cut-Off Values (μg/mL)			
	EUCAST			CLSI			EUCAST		CLSI	
Antifungals	S	ATU	R	S	I	R	WT	Non-WT	WT	Non-WT
**Isavuconazole**	≤1	2	≥4	≤1	2	≥4	≤2	≥4	≤1	≥2
**Itraconazole**	≤1	–	≥2	–	–	–	≤1	≥2	≤1	≥2
**Posaconazole**	≤0.125	0.25	≥0.5	–	–	–	≤0.25	≥0.5	≤0.25	≥0.5
**Voriconazole**	≤1	–	≥2	≤0.5	1	≥2	≤1	≥2	≤1	≥2

CLSI: Clinical Laboratory Standards Institute, EUCAST: European Committee on Antifungal Susceptibility Testing, S: Susceptible, I: Intermediate, R: Resistant, ATU: Area of Technical Uncertainty, WT: Wild Type, Non-WT: Non-Wild Type.

**Table 6 jof-11-00096-t006:** Example of a PDCA-based environmental surveillance plan for azole-resistant *Aspergillus fumigatus*.

PDCA Step	Key Activities and Considerations	Typical Timing/Frequency	Example Output/Next Step
**PLAN**	- Define main objectives (e.g., detect ≥10% resistant isolates) - Select sampling methods and sites (air sampling vs. surface swabs; identify high-risk wards, HVAC units, etc.) - Establish preliminary thresholds and alert levels (e.g., 10%) - Determine resource needs (staff, budget)	- Initial setup/revision: every 6–12 months - Triggered by major construction or seasonal changes	- Clear goals and thresholds - Resource allocation plan - Sampling schedule and staff roles
**DO**	- Implement chosen surveillance (e.g., weekly air sampling at 300 L, surface swabs) - Use selected media for Aspergillus fumigatus sense strict (e.g., Flamingo Medium) - Perform antifungal susceptibility screening to detect resistant isolates (e.g., microdilution or agar-based methods) as appropriate - Record operational details (date, location, anomalies) - Address immediate challenges (equipment calibration, staff availability)	- Weekly or monthly (routine) - Event-triggered sampling if a threshold is exceeded	- Raw data: colony counts, % resistant isolates, observational notes - Preliminary findings logged
**CHECK**	- Analyze data completeness and lab turnaround - Compare results to thresholds (e.g., is % resistance >10%?) - Identify hotspots or unexpected patterns - Review workload and cost-effectiveness - Evaluate clinical impact (e.g., correlations with IA cases)	- After each sampling cycle (e.g., monthly or quarterly review) - Additional checks if an outbreak is suspected	- Updated trend overview (resistance rates, hotspots) - Staff workload and cost review - Actionable recommendations
**ACT**	- Adjust sampling frequency or add new sites if thresholds are exceeded - Refine methods (improved media, different approach) as needed - Update alert thresholds or reporting formats - Communicate changes to relevant stakeholders (infection control team, facility managers, clinicians) - Document outcomes and lessons learned	- As soon as major issues or non-compliance are identified - Could transition into new “Plan” phase once improvements are decided	- Revised sampling plan - Enhanced cleaning/filtration - Policy changes or prophylaxis review - Document and communicate results

Abbreviations: HVAC = Heating, Ventilation, and Air Conditioning; PDCA = Plan–Do–Check–Act cycle for continuous quality improvement. This table illustrates how Plan–Do–Check–Act can guide the establishment and iterative improvement of an environmental monitoring program, including sampling methods, frequency of data review, and potential follow-up actions when thresholds are exceeded.

## Data Availability

No new data were created or analyzed in this study.

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
