# Peer review of "Comprehensive Review of Environmental Surveillance for Azole-Resistant Aspergillus fumigatus: A Practical Roadmap for Hospital Clinicians and Infection Control Teams"

_jof, 2025, doi:10.3390/jof11020096_

Round 1

Reviewer 1 Report

The manuscript "Comprehensive review of Environmental Surveillance for Azole-Resistant Aspergillus fumigatus: A Practical Roadmap for Hospital Clinicians and Infection Control Teams", presents worldwide data on the resistance of A. fumigatus to the azole and reveals a great regional variability; the authors show that the findings of other regions cannot be directly extrapolated to local environments, so mention to the need for environmental surveillance sustained at the hospital level. In addition, they describe practical approaches, which cover the prioritization of sampling sites, diagnostic workflows (based on crops and molecular tests), and continuous improvement driven by the PDCA (planning, doing, checking, and acting).

It is a well-written article that recommends linking the findings of real-time surveillance with clinical decisions so that hospitals can adapt antifungal administration programs and quickly adjust prophylaxis or treatment regimes.

Likewise, the article is interesting since the authors show the increase in the prevalence of Aspergillus fumigatus and its resistance to the azole during the last decade and propose a systematic environmental surveillance of this fungus, especially directed to detect resistant strains and provide treatment appropriate to patients. So I recommend that the article be published.

None

Reviewer 2 Report

Major comment:

This is a thorough review of the causes and consequences of an alarming and prevalent increase of resistance of Aspergillus fumigatus to azoles and other antifungals and a proposal of detailed measures to control and increase/improve the surveillance in nosocomial spaces and other environments, particularly agricultural, that are considered as the major causes of spreading of this opportunistic fungus. This organism is the etiological agent of invasive pulmonary aspergillosis (IPA) and aspergilloma in immunocompromised and highly hypersensitive individuals. I see this review as something like an “alarm cry” addressed to all community involved, in some way or another, in fungal epidemiology with emphasis on A. fumigatus and well-sustented proposals to control its worldwide spreading. Hopefully, his study reaches all community involved, in one way or another, with fungal pathogenesis and adopt all measures to control it.

Minor:

1.      Figures 1-3 are cited and legends provided but I failed to see them in the typescript.

2.      I suggest to mention in proper place of the typescript that biofilm formation by A. fumigatus is one of the most important virulence factors in IPA and aspergilloma [Kaur and Singh, 2014, Medical Mycology 52(1): 2-9.  

Major comment:

This is a thorough review of the causes and consequences of an alarming and prevalent increase of resistance of Aspergillus fumigatus to azoles and other antifungals and a proposal of detailed measures to control and increase/improve the surveillance in nosocomial spaces and other environments, particularly agricultural, that are considered as the major causes of spreading of this opportunistic fungus. This organism is the etiological agent of invasive pulmonary aspergillosis (IPA) and aspergilloma in immunocompromised and highly hypersensitive individuals. I see this review as something like an “alarm cry” addressed to all community involved, in some way or another, in fungal epidemiology with emphasis on A. fumigatus and well-sustented proposals to control its worldwide spreading. Hopefully, his study reaches all community involved, in one way or another, with fungal pathogenesis and adopt all measures to control it.

Minor:

1.      Figures 1-3 are cited and legends provided but I failed to see them in the typescript.

2.      I suggest to mention in proper place of the typescript that biofilm formation by A. fumigatus is one of the most important virulence factors in IPA and aspergilloma [Kaur and Singh, 2014, Medical Mycology 52(1): 2-9.  

Author Response

Dear Reviewer,

Thank you for your comments. We have prepared a detailed point-by-point response in the attached file. 
Please see "Reply_and_Manuscript_Tashiro_revised_ver1.0_250114_track_changes" for our full replies and the revised manuscript.

Sincerely,

Masato Tashiro

Reviewer 3 Report

The presented review is devoted to the description of the situation with the spread of azole-resistant Aspergillus strains in the world. The purpose of the review is to systematize knowledge about the spread of pathogenic and opportunistic strains of these fungi, to establish the causes of pathogenicity, to systematize practical approaches and recommended strategies for combating pathogenic fungi. The review calls for the unification of protocols for combating fungal infections worldwide to control the spread of strains and reduce the risk of infection of people worldwide. The review is very interesting, very well and clearly written. The logic of the presentation of the material is observed. The review material is fully consistent with the thematic areas of the journal. Unfortunately, the presented version of the review does not have Figures, so the review in this version is not sufficiently illustrated. Also, the text contains quite a lot of abbreviations, which makes it somewhat difficult to read.

1. Please move all Tables after their first mention in the text.

2. Table 3 should be changed to be more understable. At the least, what is in brackets should either be described in the text or highlighted in a separate column. In any case, Table 3 is difficult for the reader to understand in its current form.

3. Perhaps the "Flow of Action for Clinical and Environmental Interventions" (Section 7.2) could be presented as a diagram.

Author Response

(The authors gave the same response as above.)
